# Somatic nuclear mitochondrial DNA insertions are prevalent in the human brain and accumulate over time in fibroblasts

**Weichen Zhou**[1☉], **Kalpita R. Karan**[2☉], **Wenjin Gu**[1], **Hans-Ulrich Klein**[3,4], **Gabriel Sturm**[2,5], **Philip L. De Jager**[3,4], **David A. Bennett**[6], **Michio Hirano**[3], **Martin Picard**[2,7,8,9]*, **Ryan E. Mills**[1,10]*

1 Department of Computational Medicine and Bioinformatics, University of Michigan Medical School, Ann Arbor, Michigan, United States of America, 2 Department of Psychiatry, Division of Behavioral Medicine, Columbia University Irving Medical Center, New York, New York, United States of America, 3 Center for Translational & Computational Neuroimmunology, Department of Neurology, Columbia University Irving Medical Center, New York, New York, United States of America, 4 Taub Institute for Research on Alzheimer's Disease and the Aging Brain, Columbia University Irving Medical Center, New York, New York, United States of America, 5 Department of Biochemistry and Biophysics, University of California, San Francisco, San Francisco, California, United States of America, 6 Rush Alzheimer's Disease Center, Rush University Medical Center, Chicago, Illinois, United States of America, 7 Department of Neurology, H. Houston Merritt Center, Columbia University Translational Neuroscience Initiative, Columbia University Irving Medical Center, New York, New York, United States of America, 8 New York State Psychiatric Institute, New York, New York, United States of America, 9 Robert N Butler Columbia Aging Center, Columbia University Mailman School of Public Health, New York, New York, United States of America, 10 Department of Human Genetics, University of Michigan Medical School, Ann Arbor, Michigan, United States of America

☉ These authors contributed equally to this work.
* martin.picard@columbia.edu (MP); remills@umich.edu (REM)

**Data Availability Statement:** The Whole genome sequence of ROSMAP can be obtained through the

## Abstract

The transfer of mitochondrial DNA into the nuclear genomes of eukaryotes (Numts) has been linked to lifespan in nonhuman species and recently demonstrated to occur in rare instances from one human generation to the next. Here, we investigated numtogenesis dynamics in humans in 2 ways. First, we quantified Numts in 1,187 postmortem brain and blood samples from different individuals. Compared to circulating immune cells ($n$ = 389), postmitotic brain tissue ($n$ = 798) contained more Numts, consistent with their potential somatic accumulation. Within brain samples, we observed a 5.5-fold enrichment of somatic Numt insertions in the dorsolateral prefrontal cortex (DLPFC) compared to cerebellum samples, suggesting that brain Numts arose spontaneously during development or across the lifespan. Moreover, an increase in the number of brain Numts was linked to earlier mortality. The brains of individuals with no cognitive impairment (NCI) who died at younger ages carried approximately 2 more Numts per decade of life lost than those who lived longer. Second, we tested the dynamic transfer of Numts using a repeated-measures whole-genome sequencing design in a human fibroblast model that recapitulates several molecular hallmarks of aging. These longitudinal experiments revealed a gradual accumulation of 1 Numt every ~13 days. Numtogenesis was independent of large-scale genomic instability and unlikely driven by cell clonality. Targeted pharmacological perturbations including chronic glucocorticoid signaling or impairing mitochondrial oxidative phosphorylation (OxPhos) only modestly increased the rate of numtogenesis, whereas patient-derived *SURF1*-mutant cells

NIA Genetics of Alzheimer's Disease Data Storage Site (NIAGADS, https://dss.niagads.org/) data set NG00067 or by contacting them directly at help@niagads.org. Whole genome sequences of aged primary human dermal fibroblasts can be found in a recent study at https://columbia-picard. shinyapps.io/shinyapp-Lifespan_Study/. Illumina-sequenced 2504 independent individuals from the 1000 Genomes Project Phase 3 can be found by contacting the project at info@1000genomes.org or downloading directly from ftp.1000genomes.ebi. ac.uk/vol1/ftp/data_collections/ 1000G_2504_high_coverage/. The Numt callsets by Dinumt from 1000 Genomes Project (GRCh38) and from ROSMAP and Lifespan dataset (GRCh37), the SV callsets by DELLY, Manta, and Canvas from Lifespan dataset (GRCh37), and the MEI callset by MELT from Lifespan dataset (GRCh37) can be found at GitHub page: github. com/mills-lab/numts-and-aging-in-fibroblasts-and-brains-. The raw data for generating the plots in the figures and supplementary figures can be found in Supporting Information S1 Data. Dinumt: https:// github.com/mills-lab/dinumt. The scripts and command lines in the project can be found on the GitHub page: github.com/mills-lab/numts-and-aging-in-fibroblasts-and-brains, and ZENODO: https://zenodo.org/records/11625510 with DOI https://doi.org/10.5281/zenodo.11625510.

**Funding:** This work was supported by NIA R01AG066828 (including salary support to K.R.K., G.S., M.H., and M.P.), 1R21HG011493-01 (including salary support to R.E.M. and W.Z.), and the BaszuckiBrain Research Fund. W.Z. was partially supported by and provided salary support from the NIH/NIA-funded Michigan Alzheimer's Disease Research Center (P30AG072931) and the University of Michigan Alzheimer's Disease Center Berger Endowment. ROSMAP is supported by P30AG10161, P30AG72975, R01AG15819, R01AG17917, U01AG46152, and U01AG61356. ROSMAP resources can be requested at https:// www.radc.rush.edu. The funders had no role in study design, data collection and analysis, decision to publish, or preparation of the manuscript.

**Competing interests:** The authors have declared that no competing interests exist.

**Abbreviations:** AC, anterior caudate; AD, Alzheimer's dementia; DLPFC, dorsolateral prefrontal cortex; MAP, Memory and Aging Project; MCI, mild cognitive impairment; MEI, mobile element insertion; mtDNA, mitochondrial DNA; NCI, no cognitive impairment; OxPhos, oxidative phosphorylation; PBMC, peripheral blood mononuclear cell; PCC, posterior cingulate cortex; ROS, Religious Orders Study; SNV, single-

exhibiting mtDNA instability accumulated Numts 4.7-fold faster than healthy donors. Combined, our data document spontaneous numtogenesis in human cells and demonstrate an association between brain cortical somatic Numts and human lifespan. These findings open the possibility that mito-nuclear horizontal gene transfer among human postmitotic tissues produces functionally relevant human Numts over timescales shorter than previously assumed.

## Introduction

The incorporation of mitochondrial DNA into the nuclear genomes of organisms is an ongoing phenomenon [1–8]. These nuclear mitochondrial insertions, referred to as "Numts," have been observed in the germline of both human [6,8–11] and nonhuman [7,12–22] species. These insertions occur as part of a wider biological process termed numtogenesis [23,24], which has been defined as the occurrence of any mitochondrial DNA (mtDNA) components into the nucleus or nuclear genome. Once integrated, Numts are biparentally transmitted to future generations, like other types of genetic variation. While mostly benign, Numts have been implicated with cellular evolution and function [1,25], various cancers [23,24], and can confound studies of mitochondrial DNA heteroplasmy [26,27], maternal inheritance of mitochondria [10,28–31], and forensics [32–34].

Investigations of Numts have been conducted in numerous species, but yeast, in particular, has provided an excellent experimental platform as a model organism due to its smaller genome and fast replication timing. Mechanisms of Numt integration involve genome replication processes in several yeast species [1,2] and have further been linked to the yeast *YME1* (yeast mitochondrial escape 1) gene and double-stranded break repair [35–37]. Interestingly, Numts have also been associated with chronological aging in *Saccharomyces cerevisiae* [3], suggesting a model where the accumulation of somatic mutations with aging [38,39], particularly structural genomic changes, could provide an opportunistic environment for somatic numtogenesis.

In humans, neural progenitor cells and cortical neurons harbor extensive tissue-specific somatic mutations, including single-nucleotide variants (SNVs) [40–42], transposable elements [43–45], and larger structural variants [46–48]. However, to date, no studies have investigated the extent of Numts specific in human brain regions, though several studies have now explored somatic numtogenesis in various cancers [23,49]. Using blood as the source of DNA, rare events of germline numtogenesis leading to a new Numt absent from either parent are estimated to occur every 4,000 human births and to be more frequent in solid tumors but not hematological cancers [4]. Using the observations in yeast as a foundation, we hypothesized that the accumulation of somatic mutations with age in the human brain also could be associated with numtogenesis and an increase in the number of somatically acquired (i.e., de novo) Numts. Mechanistically, numtogenesis requires the release of mitochondrial fragments into the cytoplasm and nucleus [50,51], where they can be integrated into autosomal sequences. In this context, we note that neuroendocrine, energetic, and mitochondrial DNA maintenance stressors in human and mouse cells trigger mitochondrial DNA release into the cytoplasm [52] and even in the bloodstream [53,54]. Thus, intrinsic genetic perturbations to mitochondrial biology or environmentally induced stressors could increase numtogenesis across the lifespan.

nucleotide variant; SV, structural variation; VAF, variant allele frequency; WB, whole blood; WGS, whole genome sequencing.

We investigated these scenarios through a multifaceted approach using postmortem human brain tissue and blood from large cohorts of older individuals, as well as a longitudinal analysis of cultured primary human fibroblasts from healthy donors and patients deficient for *SURF1*, a gene associated with Leigh syndrome and cytochrome *c* oxidase deficiency [55] that alters oxidative phosphorylation (OxPhos). We further examined the potential role of environmental stress on numtogenesis through the treatment of these cells with oligomycin (OxPhos inhibitor) and dexamethasone (glucocorticoid receptor agonist).

## Results

### Somatic Numt integration differs by tissue, age, and cognitive status

We applied our Numt detection approach, *dinumt*, to whole genome sequencing (WGS) data generated in the ROSMAP cohort [56,57] comprising 466 dorsolateral prefrontal cortex (DLPFC), 260 cerebella, 68 posterior cingulate cortex (PCC), and 4 anterior caudate (AC) tissue samples as well as non-brain tissue from 366 whole blood (WB) and 23 peripheral blood mononuclear cells (PBMCs) samples (Methods, Fig 1A). We note that the sequencing coverage of these samples was too low (~45×, S1 Table) to confidently identify lower-level somatic mosaicism within any individual tissue sample, though some would still likely be detected by our approach. Our strategy instead was to examine whether there were any potential clonal mosaic events occurring predominantly in one or more tissues compared to others by filtering out all common germline Numts, with the expectation that any rare germline Numts that were not filtered out would be present at a constant rate across all tissues, age ranges, and cognitive categories and thus any differences we observe between them would be driven by somatic events.

We obtained 3,758 unique quality-pass Numt calls from ROSMAP 1,187 samples. Only 17 of these tissue samples were from the same individual, and thus typical somatic variant analysis using multiple tissues to distinguish germline and early somatic variation from tissue-specific events was prohibitive. To mitigate this, we cross-referenced all of our detected non-reference Numts against large population cohorts including the 1000 Genomes Project [6] and another recent study using the 100,000 Genomes Project in England [4]. This step identified 45 Numts shared between one of our samples and the population-level reference, which were filtered out (Methods). The remaining 3,713 Numts were then examined in aggregate across tissues to determine whether there were differences in Numt abundance between specific tissues. To identify and rectify any false positive calls that may have been triggered by potential bacterial mitochondria contamination [58], we further conducted an investigation into the length of paired-end read fragments mapped to the nuclear genome and supported Numt calls and found that in all cases (Methods), there were at least 150 bp of sequence anchored to human chromosomal sequences (S1 Fig), demonstrating that there is likely no microbiome interference in the data.

We identified a mean of 10.4 Numts per sample across all tissues, of which ~3 on average were found to be tissue-specific after filtering for germline Numt polymorphisms found in other samples, population-scale controls, or somatic Numts in other tissues, as described above. We observed no correlation between the number of Numts detected in each sample and its genomic sequence coverage ($r^2 = 0.003$, S1 Table), indicating that our results are robust across a range of sequence depths [59]. The detected Numts ranged from 22 bp to 8,172 bp in length, with a median of 73 bp, a mean of 1,169 bp, and s.d. 1,882 bp (S1 Table), consistent with previous results from population-scale data in blood DNA [4,6,7]. We observed an average of 3.52 Numts in each whole blood tissue sample, closely comparable to the 4.9 Numts in blood samples reported in Wei and colleagues [4].

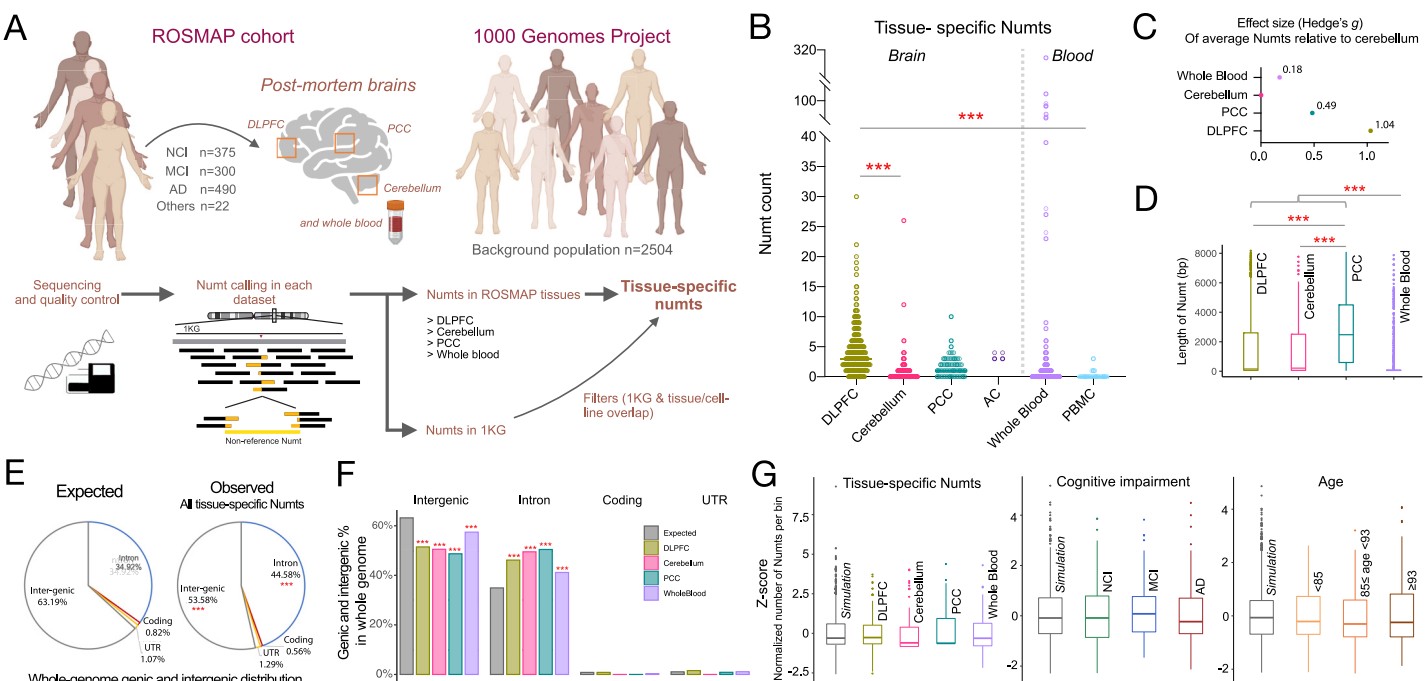

**Fig 1. Characteristics of tissue-specific Numts in postmortem human brain regions and blood.** (A) Overview of approach to identify tissue-specific Numts in ROSMAP cohort. (B) Abundance of tissue-specific Numts across brain regions and blood cells from ROSMAP participants. (C) Effect size (Hedge's *g*) of average tissue-specific Numts relative to cerebellum. (D) Length of tissue-specific Numts across brain regions and whole blood. (E) Genic and intergenic distribution of all tissue-specific Numts versus the expected distribution in the whole genome for all samples. (F) Percentage of genic and intergenic distribution of tissue-specific Numts delineated by brain regions and whole blood. (G) Random genomic distributions of tissue-specific Numts across tissues (left), cognitive impairment stratifications (middle), and age groups (right), based on comparison to simulation data. The age groups were defined as less than 85 (*n* = 295), between 85 and 93 (*n* = 545), and greater than or equal to 93 (*n* = 319). One-way ANOVA was used to test the significance in B and D. Pearson's chi-square test was used to test the significance in E. Fisher's exact test was employed instead of Pearson's chi-square test when sample sizes are small. ***, **, and * represent a significant *p*-value less than 0.001, 0.01, and 0.05, respectively. DLPFC, dorsolateral prefrontal cortex; PCC, posterior cingulate cortex. NCI, no cognitive impairment; MCI, mild cognitive impairment; AD, Alzheimer's dementia. Graphical artwork in Fig 1A were created with BioRender.com and are pursuant to BioRender's Academic License Terms. The data underlying this figure can be found in S1 Data.

Interestingly, we found the majority of tissue-specific Numts fell within DLPFC regions (mean = 4.13 per person), representing a 5.5-fold (*p*-value <0.001) higher frequency compared to the cerebellum (mean = 0.75), 2.4-fold higher than PCC (mean = 1.71, *p*-value <0.001), and 15.9-fold higher than PBMC (mean = 0.26, *p*-value <0.001) (Fig 1B). From the 17 individuals with multiple tissue samples comprising 9 pairs of the cerebellum and DLPFC samples, we consistently observed significant differences of tissue-specific Numts between the 2 tissues (S2 Fig, *p*-value = 0.033, Student's *T* test, paired, two-sided) which aligns with our larger analysis across the cohort. Relative to the cerebellum, DLPFC and AC also showed the highest effect sizes (Hedge's g = 1.03 and 1.38, respectively) in Numt abundance, followed by PCC (g = 0.49) (Fig 1C). In addition, we observed a significant correlation between the mtDNA copy number (mtDNAcn) and somatic Numts in DLPFC ($r^2$ = 0.146, *p* < 0.001), but not in other tissues (S3 Fig). mtDNAcn was higher in DLPFC than in other brain regions and tissues (median copies per cell: DLPFC = 4,047, versus cerebellum = 999, PCC = 4,042, WB = 172, and PBMCs = 125). Interestingly, we found that the length of tissue-specific Numts in 3 brain regions (DLPFC, cerebellum, and PCC) were significantly longer than those observed in whole blood (median = 63 bp, one-way ANOVA, *p*-value <0.001), with PCC-specific Numts themselves exhibiting larger lengths (median = 2,477 bp) than DLPFC and cerebellum (median = 152

bp and 210 bp, one-way ANOVA, $p$-value <0.001, Fig 1D). The absence of large Numts in blood immune cells could reflect negative selection against new Numts [4].

We next explored Numt integration using a gene-centric approach in the tissues with the largest number of samples (DLFPC, cerebellum, PCC, and WB). We focused on the Numts that were inserted in and around transcribed regions of the genome (introns, coding, UTR, and intergenic regions) and examined the proportion of our detected NUMTs that fell within each region. Surprisingly, we found that our somatic Numts integrated into introns at a significantly higher frequency compared to its overall expected proportion of the genome based on Ensembl gene annotations (44.58% v.s. 34.92%, $p$-value <0.001), while a negative enrichment was observed in intergenic regions (53.58% v.s. 63.92%, $p$-value <0.001, Methods, Fig 1E), though these could be the result of differences of sequence mappability within these regions precluding the detection of Numts. These significant differences in genomic distributions were further observed across the various tissues (Fig 1F).

We lastly hypothesized that the genomic distribution of somatic Numt integration sites might differ within individual tissues, cognitive status, or age groups from an expected random distribution throughout the genome. We tested this hypothesis by iteratively assigning random positions for each of our observed Numts 50,000 times across the human genome reference and assessing whether our observed integration sites differed significantly when compared within predefined 10 Mb windows across the genome as a permutation test. After the multiple test correction (Benjamini–Hochberg procedure), we observed no significant difference from random for any of the tested tissues (Fig 1G). We further stratified our results by age and cognitive status (Methods) and likewise observed no differences in genomic distribution. This is in agreement with previous studies that suggest Numt integration is a random occurrence [6,7,11], though we note that the paucity of Numts may lead to the permutation test being underpowered and thus preclude an accurate assessment at such broad regions across the genome.

## DLPFC-specific Numts are negatively associated with age at death in persons without cognitive impairment

On the basis of potential adverse genomic effects of Numts [4,50,51] and the results above, we hypothesized that tissue-specific Numts were associated with mortality and age at death, though this correlation might differ between tissues or clinical diagnoses. We first examined NUMTs in aggregate across tissues and observed almost no correlation in Numt abundance with the age of death (Fig 2A). Given the existence of mitochondrial DNA defects alterations in the human brain with cognitive decline [60], we next stratified individuals with tissue-specific Numts by their cognitive status into no cognitive impairment (NCI), mild cognitive impairment (MCI), and Alzheimer's dementia (AD, COGDX score, Methods, S1 and S2 Tables). We found that in DLPFC tissues, NCI individuals that carried more Numts died earlier ($r^2 = 0.094$, $p$-value <0.001), with 2 additional Numt insertions observed per decade of life lost (Fig 2B). MCI individuals exhibited a similar but lower negative association ($r^2 = 0.031$, $p$-value <0.05). However, no correlation between Numts and age at death was observed in the AD group ($r^2 = 0.009$, $p$-value = 0.19). In the cerebellum, we observed similar patterns of correlation between Numts and age at death among cognitive groups, albeit with weaker correlations (NCI: $r^2 = 0.044$, $p$-value <0.05; MCI: $r^2 = 0.007$, $p$-value = 0.514; and AD: $r^2 = 0.045$, $p$-value <0.05, S4 Fig). These results indicate that Numts are negatively associated with age at death in certain brain regions of non-AD individuals, suggesting the possibility that brain numtogenesis is deleterious and that the pathogenicity of AD may be uncoupled from age-dependent Numt integration.

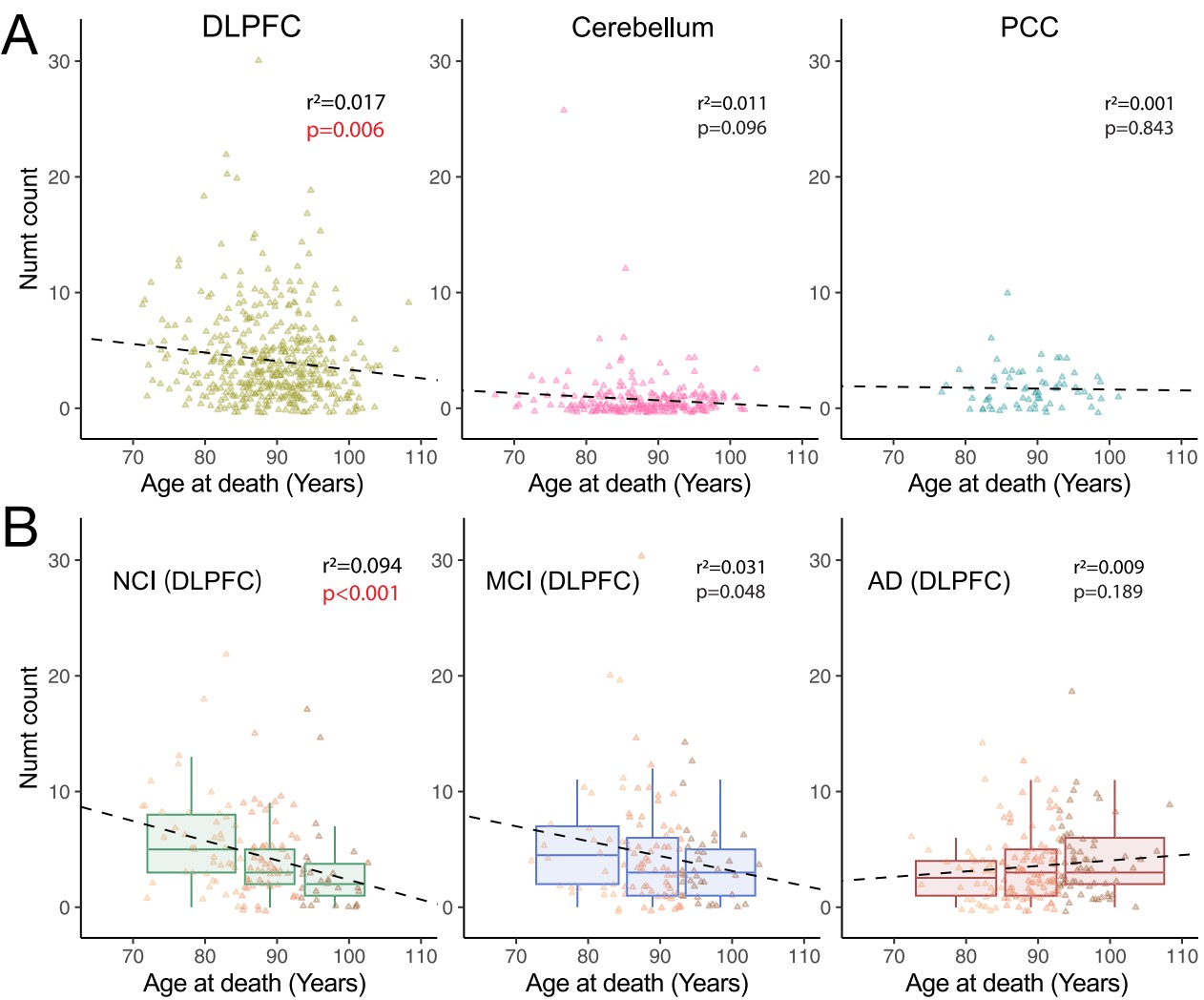

**Fig 2. Numt association with age at death by tissue and cognitive impairment.** (A) Correlation between age at death and abundance of DLPFC-specific, cerebellum-specific, and PCC-specific Numts, respectively. (B) DLPFC samples correlated with age at death, stratified by cognitive diagnosis status as NCI ($n = 121$, left), MCI ($n = 112$, middle), and AD ($n = 176$, right). Data points are colored by arbitrary age groups (see Methods) in light yellow, orange, and brown, respectively. $r^2$ and $p$-values are calculated using standard least-squares regression models. DLPFC, dorsolateral prefrontal cortex; PCC, posterior cingulate cortex. NCI, no cognitive impairment; MCI, mild cognitive impairment; AD, Alzheimer's dementia. The data underlying this figure can be found in S1 Data.

## Somatic Numts accumulate over time in fibroblasts

The cross-tissue analysis of the ROSMAP cohort provided compelling results that Numts have both tissue and age-dependent characteristics of their integration in aggregate. However, the lack of relationships between individual tissue samples prohibits a direct measurement of Numt integration rates or numtogenesis. We therefore tested the dynamic transfer of Numts using a longitudinal, repeated-measures WGS study in primary human fibroblasts cultured in vitro under physiological conditions (Table 1) [5,61–63]. Over time, replicating cells exhibit conserved epigenomic (hypomethylation), telomeric (shortening), transcriptional (senescence-associated markers), and secretory (pro-inflammatory) features of human aging, representing a useful model to quantify the rate of dynamic age-related molecular processes in a human system [5]. We recently showed that primary mitochondrial bioenergetic defects

**Table 1. Experimental design for Numt accumulation study in the in vitro fibroblast aging model.**

| Individual | Gender | Treatment | Collection day | # Data point |
|---|---|---|---|---|
| Control | | | | |
| Donor 1 | Male | None | 7 ~ 211 | 8 |
| | | Dex | 27 ~ 166 | 8 |
| | | Oligo | 27 ~ 103 | 6 |
| Donor 2 | Female | None | 8 ~ 189 | 8 |
| | | Dex | 43 ~ 211 | 10 |
| | | Oligo | 28 ~ 189 | 8 |
| Donor 3 | Male | None | 3 ~ 189 | 10 |
| | | Dex | 28 ~ 90 | 4 |
| | | Oligo | 28 ~ 105 | 4 |
| *SURF1* defect | | | | |
| Patient 1 | Male | None | 16 ~ 90 | 5 |
| Patient 2 | Male | | 8 ~ 152 | 8 |
| Patient 3 | Female | | 8 ~ 58 | 5 |

accelerate the rate of aging based on the telomere shortening per cell division, DNA methylation clocks, and age-related secreted proteins [63]. Therefore, using this model to monitor the accumulation of overall and donor-specific unique Numts absent in the general population, we analyzed cultured fibroblasts from 3 unrelated healthy donors, aged in culture under physiological conditions for up to 211 days [5] (Fig 3A). Instead of focusing on a single cell line tested in triplicates, we opted to include 3 separate donors, which provides a more robust test of our hypothesis.

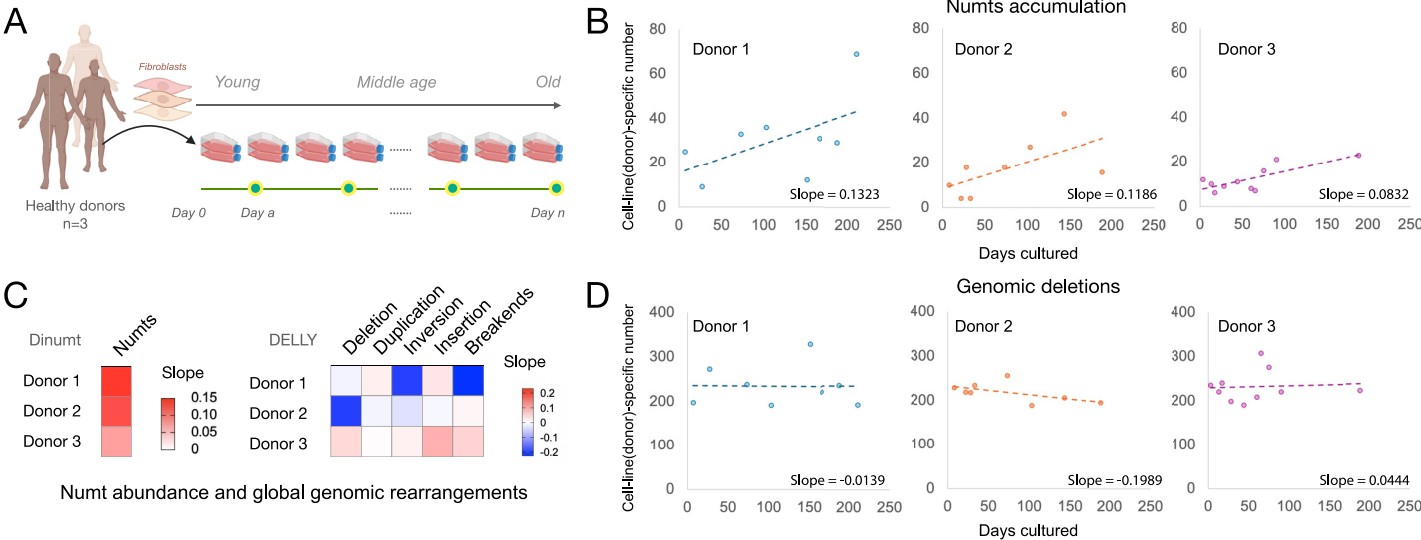

**Fig 3. Numts accumulate in human fibroblasts during normal aging.** (A) Study design of cellular aging model using primary fibroblasts from 3 healthy donors. (B) Cell line-specific Numts accumulate over time in aging fibroblasts obtained from 3 healthy donors cultured up to 211 days. (C) Heatmap of slopes based on the linear regression between days cultured and the cell line-specific calls, including Numts (from Dinumt, left) and structural variants (from DELLY, right). (D) Time-course of cell line-specific deletions in the 3 healthy donors. Graphical artwork in Fig 3A were created with BioRender.com and are pursuant to BioRender's Academic License Terms. The data underlying this figure can be found in S1 Data.

Across the 3 donors, we observed a positive correlation between time in culture and the number of unique Numts ($r^2$ = 0.30, 0.38, and 0.59, respectively) with a positive slope range of 0.08 to 0.13 Numts/day. Thus, on average, human fibroblasts accumulated a novel Numt every 12.6 days of culture (or 0.79 Numt per 10 days, 95% C.I. = 0.28 to 1.31). Numtogenesis was also evident when delineating our Numts per donor into total Numts observed at each time point (total Numts) and Numt insertions specific to individual cell lines with treatment within each donor (cell line-specific Numts) (Figs 3B and S5).

To determine if global genomic instability could account for this effect, we conducted the same analysis on multiple types of somatic structural variation (e.g., deletion, duplication, inversion, insertion, and breakends; Methods). The accumulation rates of these variant types were significantly lower compared to the rate of numtogenesis (Fig 3C). For example, although we observed a positive yet more moderate increase in deletion abundance compared to Numts when considering all such variants, we did not observe the same increase in the cell line-specific deletions as with the somatic Numts (Fig 3D). These results indicate that Numt insertions occur at a higher rate than autosomal deletions in this system and suggest a higher rate of age-related somatic numtogenesis rate than the other genetic variants (S6 Fig). We further observed no significant increase in cell line-specific genomic duplications over time, thus indicating that the increase in the total number of Numts over time is likely due to novel integration events and not duplications of preexisting copies.

We questioned whether the accumulation of these apparently somatic Numts could be driven by the simple clonal expansion of few Numts-containing cells. Even within a given donor line followed longitudinally, all observed Numts were unique in their length and sequence (average 562 bp, s.d. 1,400 bp) and showed no evidence of relatedness with one another. This lack of sequence overlap is most parsimoniously explained by the random nature of our sequencing coverage (sequencing depth: 25×; the total number of genomes in each experiment $2 \times 10^6$ diploid genomes) and a large number of new Numts accumulating over time. Thus, the unique identity of all observed Numts in these in vitro experiments argues against the clonal origin of these events.

## Impact of environmental and genetic stress on somatic Numt integration rates

We next explored whether the cellular environment could impact somatic numtogenesis by testing if chronic exposure to a stress-mimetic or an inhibitor of mitochondrial OxPhos would alter the rate of Numt accumulation in otherwise healthy and aging fibroblasts. We analyzed human fibroblasts derived from the same 3 healthy donors described above that were treated with (a) the glucocorticoid receptor agonist dexamethasone (Dex, 100 nM); and (b) the ATP synthesis inhibitor oligomycin (Oligo, 100 nM). Similar to the untreated donors (see Methods, $r^2$ = 0.30, *p*-value = 0.004 linear regression), both treatment groups exhibited an accumulation of new Numts over time (see Methods, linear regression for Dex group $r^2$ = 0.59, *p*-value <0.001; for Oligo group $r^2$ = 0.22, *p*-value = 0.052). Compared to the untreated group (0.79 Numt per 10 days, 95% C.I. = 0.28 to 1.31), Dex and Oligo treatments tended to increase the rate of numtogenesis to 1.07 Numt (95% C.I. = 0.65 to 1.49) and 2.15 Numt (95% C.I. = −0.02 to 3.27) per 10 days, respectively (Fig 4A, 4B and 4D). Although these differences in effects did not reach statistical significance, the accumulation of Numts over time in these biologically independent experiments from those above further document Numtogenesis in aging human cells in vitro.

Using a genetic approach, we further tested whether defects in mitochondrial OxPhos associated with mtDNA instability are sufficient to alter the rate of numtogenesis. We analyzed

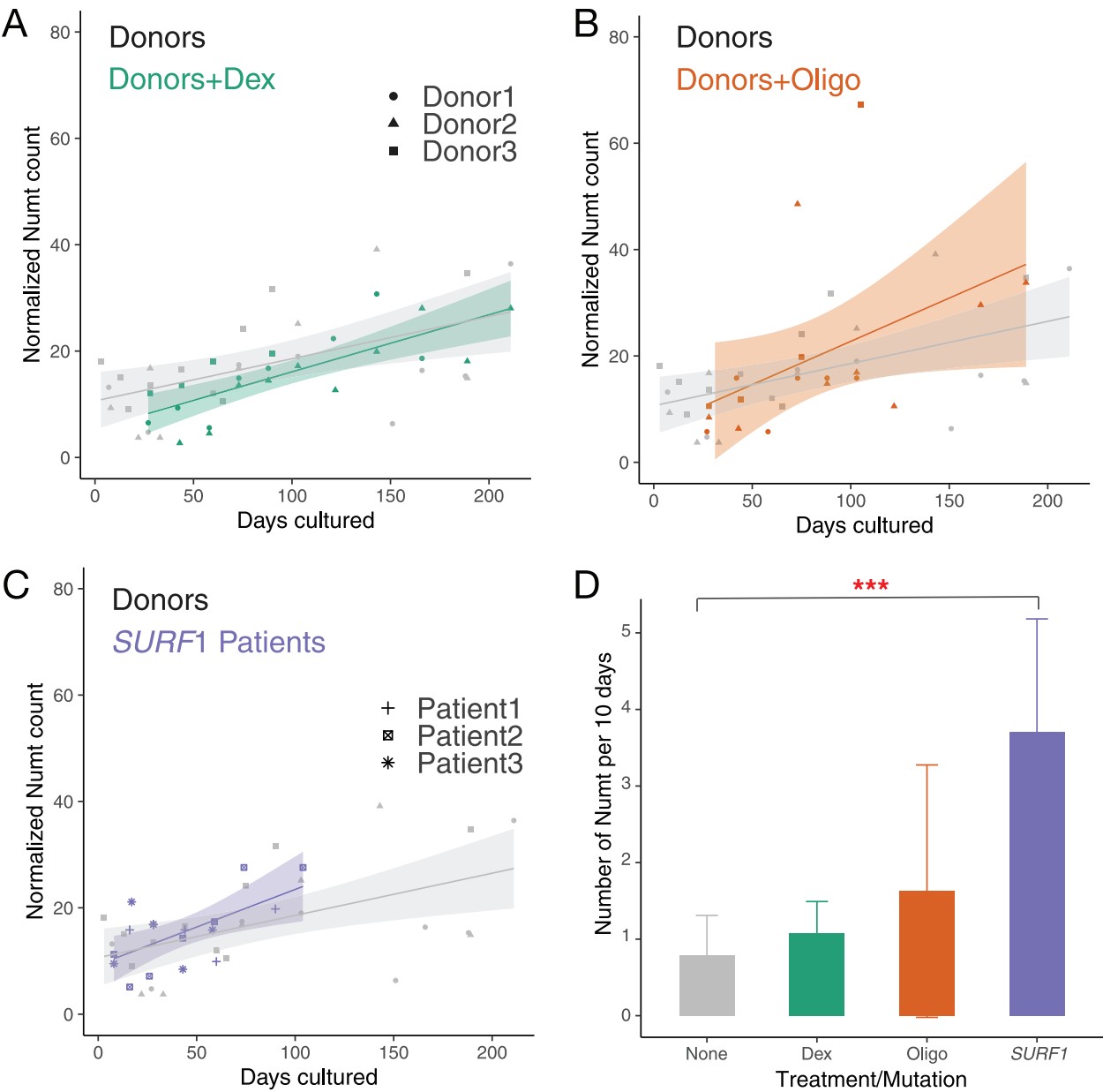

**Fig 4. Effect of chronic pharmacological and genetic perturbation of mitochondrial function on Numt accumulation in fibroblasts.** (A) Time course of Numt accumulation in healthy donors 1–3 and same cells cultured in dexamethasone (Dex) mimicking chronic Dex exposure. (B) Time course of Numt accumulation in healthy donors 1–3 and same cells cultured in oligomycin (Oligo). (C) Numt accumulation time course in the patient fibroblasts with *SURF1* gene defect (Patient 1–3) and the ones from 3 healthy donors (Donor 1–3). (D) Comparison of Slopes derived from all patients untreated, Dex-treated, oligo-treated, and *SURF1* gene defect. Numt counts for each group were normalized by the median value. A linear regression analysis was performed to derive the rate of the Numt accumulation and calculate the slopes in each group. ANOVA test was used to test the significance between the slopes of untreated donors and the ones of treated donors or patients in the hypothesis that pharmacological or genetic perturbation would increase the accumulation of Numts. \*\*\*, \*\*, and \* represent a significant *p*-value less than 0.001, 0.01, and 0.05, respectively, and ns represents non-significance. The data underlying this figure can be found in S1 Data.

data from a similar fibroblast culture system in 3 patient-derived fibroblasts with *SURF1* mutations (Patient 1–3) [5]. Mutations in *SURF1* represent one of the most frequent causes of cytochrome c oxidase and OxPhos deficiency in humans [64], and we recently showed that these *SURF1*-mutant fibroblasts accumulate large-scale mtDNA deletions over time in culture,

demonstrating mtDNA instability in this model. In these independent donors, we again observed the accumulation of new Numts over time (regression $r^2 = 0.64$, *p*-value <0.001). Strikingly, the rate of Numtogenesis in *SURF1*-mutant cells was 3.71 Numt per 10 days (95% C.I. = 2.24 to 5.18), in contrast to the rate of 0.79 Numt per 10 days in the healthy donors (4.7-fold of control, *p*-value = 8.20E-05, ANOVA test, Fig 4C and 4D). As in tumors [23,24] and yeast [1,2], this result further documents Numtogenesis as an inter-genomic event occurring in human cells over relatively short timescales, and establishes, using patient-derived cells with mtDNA instability, the modifiability of the rate of numtogenesis in vitro [63].

## Discussion

The transfer of mitochondrial DNA into the nuclear genome of eukaryotes occurs in the germlines of various eukaryotes, suggesting that the endosymbiotic event initiated 1.5 billion years ago is still ongoing [4]. However, the extent and impact of somatic Numt insertions in specific human tissues have remained elusive outside of cancerous environments. Here, we provide some of the first evidence of the somatic nuclear accumulation of Numts in both healthy and impaired human brain tissue across different age ranges. We found that specific brain regions harbor more somatic insertions than others, in line with previous studies of other types of genomic variation [46–48], and that these rates do differ with the degree of cognitive impairment. We further extend these observations in a longitudinal study of primary human fibroblasts under various environmental and genetic conditions, documenting variable rates of ongoing numtogenesis in dividing human cells.

Our data provide new information concerning the rate of numtogenesis in humans. While the cross-sectional study of postmortem human brains does not allow us to draw conclusions about the rate of Numt transfer, the higher frequency of Numts in postmitotic brain tissue—relative to the commonly studied genomic material from blood—indicates a greater number of Numts. We further observed a significant correlation with mtDNA copy number in these samples, suggesting a potential biological mass action mechanism whereby higher mtDNAcn results in more potential transfers to the nuclear genome (S3 and S7 Figs). On the other hand, our longitudinal in vitro studies allow us to measure Numts among the same cell population (i.e., "individual") over time as cells accumulate age-related molecular hallmarks of aging. In this system, cells divide approximately every 40 h (1.7 days) when they are young (from days 0 to 80) and slow their replication rate dramatically towards the end of life when they undergo less than 1 replicative event per month (see growth curves in [65]). Although the temporal resolution of our trajectories is limited by the number of time points across the lifespan of each cell line (on average 7), our data suggests that the rate of numtogenesis is roughly linear across the wide range of replication rate, and therefore, more dependent on time rather than the rate of replication. This observation aligns with the results in cortical brain tissue (DLPFC), a postmitotic tissue where cell division is expected to be minimal to absent. This result is unlikely explained by clonality as all discovered Numts are unique. Consistent with the ROSMAP tissue data suggesting the accumulation of somatic Numts detected at very low variant allele frequency (VAF) in genomic material (median = 4.4%, mean = 5.3% ± 3.5%, with 99.7% of total number VAF <35%, S8 Fig), the VAF distribution for somatic Numts in fibroblasts was comparably low (median = 9.1%, mean = 13.1% ± 7.4%, with 98.3% of total number VAF <35%, S8 Fig). In cultured fibroblasts, each cell population contains between 1 and 5 million cells, meaning 2 to 10 million genomic copies. At 25× coverage, we therefore sample 0.00125% to 0.00025% of all copies. Our sampling is therefore essentially random. If the accumulation of Numts in aging fibroblasts was driven by clonality, we would see an increase in abundance of a

few Numts present at each passage in the same donor, rather than an increased frequency of unique Numts at each time point, as observed here.

At least 2 main findings from our results and that of others suggest that de novo Numts may be functionally meaningful. First, strong evidence that detectable Numts are excluded from coding DNA sequences and instead preferentially integrate within intergenic regions [4,6–8] suggest that they are under some functional constraint during development. In contrast, tumors frequently contain Numts within genes, and Numts may even contribute to oncogenesis, which in this case would drive positive selection in tumors [4]. Numts have also been implicated previously in several diverse disorders [66–71]. Second, Numts prevalence correlates with the somatic selection pressure for adverse mitochondrial genomic changes. In the replicating blood immune compartment of the bone marrow, high selection pressure occurs and eliminates mtDNA mutations over time [72,73], whereas de novo mtDNA defects accumulate at high levels in postmitotic tissues such as skeletal muscle, heart, and in the brain [74]. Similarly, it is possible that Numt insertions which negatively affect cell fitness in the lymphoid or myeloid lineages of the bone marrow are outcompeted and eliminated from the cell pool (or exist at low levels and are not sampled during blood draw), compared to somatic tissues where the same potentially deleterious Numt insertions in postmitotic cells cannot readily be outcompeted and eliminated without functionally compromising the tissue.

Some limitations of our study should be noted. While our human multi-tissue study includes >1,000 individuals, given the low number of Numts per person, we are likely underpowered to draw definitive conclusions around the cartography of Numts, such as potential Numts hotspots in the nuclear genome. The report by Wei and colleagues in 66,083 human genomes robustly addresses this point [4]. The absence of WGS data from multiple tissues in the same individuals also precludes direct comparisons of the specific rate of numtogenesis between tissues. This question could be addressed in other studies (e.g., GTEx) by sequencing dozens of tissues from the same individuals, but the current lack of such data set precludes a robust analysis of this kind. In relation to our longitudinal cellular studies with genetic and pharmacological mitochondrial OxPhos defects, the marginal increase in cell death when OxPhos is disrupted [63] or with chronic glucocorticoid stress [75] could offer a route of negative selection that eliminates deleterious de novo Numts; the most compromised cells die, cleansing the cell population of new variants. If this effect was strong, it would make numtogenesis undetectable to WGS, or reduce the observed effect size of the rate of numtogenesis in both our brain and fibroblast studies. This possibility could be addressed in future studies by systematically sequencing cellular debris or dead cells from the culture medium. Thus, while our positive results conclusively establish the dynamic nature of Numts transfer in healthy and stressed human cells, the magnitude reported across donors (range of 0.79 [controls] to 3.71 [*SURF1*-mutant] Numts every 10 days) may reflect a lower bound, and the rates of numtogenesis in cells under stress should be interpreted with caution. Validating the dynamic nature of numtogenesis across the lifespan in humans would require repeated measures, longitudinal WGS of postmitotic tissues (e.g., muscle biopsies), with the caveat that the same exact cells likely cannot be repeatedly biopsied, and therefore that new Numts would be missed. For the reasons mentioned above (negative selection in the immune compartment), repeated WGS in blood would likely underestimate the true rate of in vivo mito-nuclear genomic transfer.

We also considered if the increase in Numts over time could be driven by the clonal expansion of Numt-containing cells, which would suggest that there is no active transfer of mtDNA and numtogenesis in this model. Here, our observed increase in unique Numts over time suggests that these are not the product of clonal expansion, which also is consistent with the somatic Numts in ROSMAP samples, and in Wei and colleagues [4]. Several possible theories have been advanced for numtogenesis, with several studies showing environmental effects of

ionizing radiation [76,77] or mitochondrial reactive oxygen species [3,78] causing double-strand breaks associated with Numt accumulation. However, there have been few mechanistic studies in this area to conclusively determine the precise process. What is clear is that for such events to happen, whole or fragments of mtDNA must come in close proximity to the autosomal genes. There are now several reports of cytoplasmic mtDNA release where the mtDNA is released in "free form" and able to bind DNA-sensing receptors such as cGAS [50,79–81]. In fact, partial mitochondrial permeabilization that may lead to cytoplasmic mtDNA release triggers nuclear genomic instability [82], which could theoretically open the door ongoing to numtogenesis, possibly independent of cell division and genome replication.

In conclusion, our results demonstrating the high prevalence of non-germline Numts in hundreds of human brains and their negative association with age at death suggests that numtogenesis occurs across the human lifespan and that they may have deleterious health effects. Using a longitudinal in vitro human system, we establish that primary human fibroblasts accumulate Numts over time and that numtogenesis may be accelerated by some stressors, in particular, *SURF1* defects associated with mtDNA instability. These findings build and extend previous evidence that numtogenesis is active in the human germline and can have deleterious genomic, cellular, and health effects on the host organism. The active transfer of mtDNA sequences to the nuclear genome adds to the vast repertoire of mito-nuclear communication mechanisms [83] that shape human health.

## Materials and methods

### Ethics statement

The ROS and MAP studies were approved by the Institutional Review Board of Rush University Medical Center, protocols #L91020181 (ROS), L86121802 (MAP), and #L99032481 (RADC repository). All participants signed an informed consent, Anatomical Gift Act, and a repository consent to share data and biospecimens. Patient-derived fibroblasts were utilized from the previous study and were approved through the Columbia University Irving Medical Center IRB #AAAB0483 in a previous publication [5]. All human studies were conducted according to the principles expressed in the Declaration of Helsinki.

### ROSMAP cohort

**Study participants.**   The Rush Memory and Aging Project (MAP) and the Religious Orders Study (ROS) [56,57] are 2 ongoing cohort studies of older persons, collectively referred to as ROSMAP. The ROS study enrolls older Catholic nuns, priests, and brothers, from more than 40 groups across the United States. The MAP study enrolls participants primarily from retirement communities throughout northeastern Illinois. Participants in both cohorts were without known dementia at study enrolment and agreed to annual evaluations and brain donation on death.

The clinical diagnosis of AD proximate to death was based on the review of the annual clinical diagnosis of dementia and its causes by the study neurologist blinded for postmortem data. Postmortem Alzheimer's disease pathology was assessed as described previously [84,85] and Alzheimer's disease classification was defined based on the National Institutes of Ageing-Reagan criteria [86]. Dementia status was coded as NCI, MCI, or AD from the final clinical diagnosis of dementia and the NIA Reagan criteria as previously described [87–89].

**Data processing.**   We obtained and processed WGS samples from these cohorts through the NIA Genetics of Alzheimer's Disease Data Storage Site (NIAGADS) data set NG00067. In brief, we obtained sequence data for 1,187 tissue samples comprising 466 DLPFC, 260 cerebella, 68 PCC, 4 AC, 366 WB samples, and 23 PBMCs. We obtained these 1,187 samples from

1,170 individuals, where only 17 participants in the data set contributed 2 tissue different samples to the sequencing data (S1 and S3 Tables). All sequencing data were provided in CRAM format and aligned to the human genome reference (GRCh37) with an average read depth of 45×. Based on the clinical information of age, we also stratified all 1,187 samples into 3 age groups with a roughly similar sample size based on their percentiles (30%ile, 40%ile, and 30%ile) when fit to a Gaussian distribution: (a) samples died at an age younger than 85; (b) age at death older than or equal to 85, and less than 93; and (c) age at death older than or equal to 93 (S2 Table). All the alignment statistics are presented in S1 Table, along with the clinical characteristics of the study participants.

### In vitro fibroblast aging model

**Fibroblast collection and passaging.** We further made use of processed WGS generated from a recent study of aged primary human dermal fibroblasts [5]. In brief, primary human dermal fibroblasts were obtained from distributors or our local clinic from 3 healthy and 3 *SURF1*-patient donors. Fibroblasts were isolated from biopsy tissue using standard procedures. Cells were passaged approximately every 5 days (+/− 1 day). Study measurements and treatment began after 15-day culture to allow for adjustment to the in vitro environment. Treatment conditions for healthy controls include the chronic addition of 1 nM oligomycin (oligo) to inhibit the OxPhos FoF1 ATP synthase and 100 nM dexamethasone (Dex) to stimulate the glucocorticoid receptor as a model of chronic stress [75,90]. Time points collected vary by assay, with an average sampling frequency of 15 days and 4 to 10 time points for each cell line and condition. Individual cell lines were terminated after exhibiting less than 1 population doubling over a 30-day period, as described in [5].

**Whole-genome sequencing and processing.** Whole-genome sequencing data were performed in the lifespan samples at each time point (overall 85 time points). Paired-end reads were aligned to the human genome (GRCh37) using Isaac (Isaac-04.17.06.15) [91]. Samtools (Ver1.2) [92] and Picard Toolkit (https://broadinstitute.github.io/picard/) were further used to process the aligned bam files and mark duplicates. The average read depth from the WGS and other alignment statistics used in this study can be found in S4 Table.

### Population-scale WGS control data

We leveraged 2,504 independent individuals from the 1000 Genomes Project Phase 3 to serve as population-level controls. Samples were sequenced by 30× Illumina NovaSeq (https://ftp.1000genomes.ebi.ac.uk/vol1/ftp/data_collections/1000G_2504_high_coverage/) [93], and the data were archived in CRAM format with the GRCh38 reference (https://ftp-trace.ncbi.nih.gov/1000genomes/ftp/technical/reference/GRCh38_reference_genome/). We also used Numts reported in the 100,000 Genomes Project in England [4] as population-level controls.

### Detection of non-reference Numts

We applied an updated version of Dinumt [6,7,94] to identify non-reference Numts across the different sequenced cohorts. Dinumt is an established software that was first used in the 1000 Genomes Project and validated by orthogonal methods, including PCR and Sanger sequencing [6], and long-read sequencing [7]. Briefly, Dinumt identifies aberrant/discordant reads aligning with either the mtDNA or the reference Numts on one end and map elsewhere in the genome on the other end, read orientation, and various other filters to define insertion breakpoints. Reads are discarded if they do not align uniquely to the nuclear genome and have a mapping quality (MAPQ) of less than 10. Identified insertions are then filtered for quality using a Phred scale (≥50), a cutoff of supporting reads (≥4), and a cutoff of read depth (≥5×)

around the insertion point. We built the first set of Numts as populational and polymorphic controls from the 2,504 individuals of the 1KG Project recently re-sequenced to high coverage [93]. Individual non-reference Numt callsets were resolved and merged into a single VCF file using the merging module of Dinumt. All "PASS" Numts are lifted over [95,96] from GRCh38 to GRCh37 for the downstream analysis. Dinumt was used to identify non-reference Numts across the individual sequences from 1,187 samples in ROSMAP or 85 cell line genomes in the lifespan model. The same criteria were conducted in the pipelines (Fig 1A). The VAF of each Numt call was calculated based on the number of supporting reads reported by Dinumt divided by the number of overall read coverage in the sequenced genome. In addition, the length distribution of paired-end read fragments mapped to the nuclear genome and supported Numt calls were calculated to investigate the potential bacterial mitochondria contamination.

We then cross-referenced all of our detected non-reference Numts against large population cohorts including the 1000 Genomes Project and Numts reported from 66,083 genomes in the 100,000 Genomes Project in England [4]. In each case, we considered a Numt detected in our analysis as a germline polymorphic insertion if it fell within +/− 50 bp of a Numt reported in either of these studies.

We identified tissue/cell line-specific Numt insertions using the identified non-reference Numts. Tissue-specific Numts were derived from the ROSMAP callset, and cell line-specific calls were derived from the Numt callset of fibroblast lifespan data. Numts from each sample were first merged into an aggregated set for these 2 callsets. We then extracted all non-reference Numts that were found in only 1 specific tissue across all samples or cell line, respectively (S1 and S4 Tables). All analysis pipelines and the command lines for running Dinumt can be found at https://github.com/mills-lab/numts-and-aging-in-fibroblasts-and-brains [97].

## Statistical analysis

Cell line-specific Numts from fibroblast lifespan data were grouped by both donor/patient and treatment status. There are 4 treatment statuses: no treatment donors, donors/cells cultured in dexamethasone (Dex), donors/cells cultured in oligomycin (Oligo), and patients' fibroblasts with *SURF1* gene mutation. To increase the statistical power of data points in each group, we normalized the Numt count, merged the data points from individual samples, and then conducted the linear regression. Normalized Numts counts were derived by normalizing by the median of Numt counts in each group as defined above for various donor/patient and treatment status combinations. A linear regression model was constructed for each category respectively as below:

$$Y = \beta_0 + \beta_1 X + \epsilon$$

$$Y : \textit{Normalized Numts Count}; \ X : \textit{Days cultured}$$

Slopes ($\beta_1$) were compared by ANOVA test separately between 2 categories. All statistical analyses were performed in R 4.0.5.

## Genomic analyses for non-reference Numts

**Numt hotspots across chromosomes.** We delineated the entire nuclear genome into 10 Mbp bins, resulting in an average of 10 detected Numts per bin. The frequency for tissue-specific Numts from ROSMAP in each bin was calculated. We performed a permutation analysis by randomly shuffling the genomic positions of each observed Numt 50,000 times to determine any hotspots across genome bins compared to the real data. An empirical *p*-value was

calculated for all the bins based on the frequency of Numt ranking in simulation data. A multiple test correction (Benjamini–Hochberg) was further conducted to decrease the false discovery rate. A bin with a *p*-value less than 0.05 after the adjustment was defined as a significant hotspot. A Z-score is calculated for normalizing Numt count, which measures the deviation of the Numt count in each 10 Mbp bin from the genome-wide average across all the bins. We stratified the tissue-specific Numts into different tissues, cognitive impairment levels, and age groups to perform the hotspot analysis separately.

**Genomic content analysis and functional annotation.** We conducted the genomic content analysis for non-reference Numt insertions. We calculated the genic distribution for the tissue-specific Numts from ROSMAP. Gene track (GRCh37) was obtained from Ensembl Genome Browser (https://grch37.ensembl.org/). Parameters for protein-coding regions, transcriptomes, and exons were calculated based on a previous report [98]. Pearson's chi-square test and Fisher's exact test were used to assess the statistical significance of the discrepancies between the observed and expected distributions of Numts within genic or intergenic regions. We compared 2 × 2 categorical variables, specifically the proportion of Numts within a particular genomic region against the one in the rest of the genome, with respect to either the observed or expected data. GC content and repeat sequence analyses were carried out both in the set of cell line-specific Numt insertions from the lifespan model and polymorphic Numts from the 1000 Genomes project. GC content and repeat sequence were downloaded from the GC content table and RepeatMasker track in the UCSC Genome Browser (https://genome.ucsc.edu/). Gene mapping was carried out by AnnotSV (https://lbgi.fr/AnnotSV/) [99] to determine the genes that were potentially affected by the tissue-specific Numts from ROSMAP (S5 Table) or cell line-specific Numts from the lifespan model (S6 Table).

## Detection of structural variation (SV)

Background structural variations (SVs) were detected in the data of the 1000 Genomes Project, ROSMAP, and lifespan model. We used an integrated non-reference SV callset from the 1000 Genomes Project as the control in the project to filter out potential non-somatic SVs at the population level. It was derived from 13 callers and can be obtained from http://ftp.1000genomes.ebi.ac.uk/vol1/ftp/data_collections/1000G_2504_high_coverage_SV/working/20210104_JAX_Integration_13callers/. Delly2 (Version 0.8.5) [100] was applied to resolve non-reference SVs (including deletions, duplications, insertions, inversions, and translocations), and MELT (Version 2.1.4) [101] was used to identify a specific type of non-reference SVs, mobile element insertions (MEIs, including Alus, LINE-1s, and SVAs), in the sequenced genomes of lifespan experiments. Manta [102] and Canvas (Version 1.28.0) [103] were also applied to resolve non-reference SVs in the sequenced genomes of lifespan experiments. The same pipeline used in Numts was implemented to identify tissue-specific or cell line-specific SVs/MEIs among the ROSMAP and lifespan samples (S1 and S4 Tables).

## Mitochondrial DNA copy number

We calculated the mtDNAcn using autosomal coverage with the following formula: $mtDNAcn = (cov_{MT} / cov_{autosomal}) \times 2$ [60] in both the ROSMAP data set and lifespan study. The median sequence coverages of the autosomal chromosomes *covnuc* and of the mitochondrial genome *covmt* were calculated using R/Bioconductor (packages GenomicAlignments and GenomicRanges). We filtered out the reads with MAPQ = 0 in the analysis. Ambiguous regions were excluded using the intra-contig ambiguity mask from the BSgenome package. The mtDNAcn was z-standardized within each brain region and DNA extraction kit and then logarithmized. The normalization facilitated the combined analysis of the 2 different kits used for the DLPFC

and resulted in approximately normal mtDNAcn measures [60]. To note, using either the median or mode will not have a noticeable impact on the mtDNA copy number estimates and downstream analyses (S9 Fig). R and shell scripts used for mtDNA analysis are deposited at GitHub: https://github.com/cu-ctcn/mtDNA [60].

Cellular mtDNA content was also quantified by qPCR on the same genomic material used for other DNA-based measurements in the ROSMAP and lifespan studies. Duplex qPCR reactions were performed to simultaneously quantify mitochondrial (mtDNA, ND1) and nuclear (nDNA, B2M) amplicons, details of which can be found in the previous studies [5,104]. We observed a significant correlation between qPCR and WGS method in terms of mtDNAcn. Spearman correlation analysis between mtDNAcn measures from qPCR versus WGS is presented across cell lines by treatment (S10 Fig).

## Supporting information

**S1 Table. Meta table for ROSMAP sequencing data, including alignment statistics, clinical information, and variant numbers.**
(XLSX)

**S2 Table. Sample counts in different age groups (age at death) and cognitive status across main tissues in ROSMAP.**
(XLSX)

**S3 Table. The list of 17 participants contributing 2 tissue samples in the ROSMAP with sample ID, information, and numbers of detected somatic Numt.**
(XLSX)

**S4 Table. Meta table for lifespan sequencing data, including alignment statistics, experimental information, and variant numbers.**
(XLSX)

**S5 Table. Meta table for gene annotation of tissue-specific Numts and Numt callset from ROSMAP.**
(XLSX)

**S6 Table. Meta table for gene annotation for cell line-specific Numts and Numt callset from the lifespan model.**
(XLSX)

**S1 Fig. Length distribution of the fragments in paired-end reads mapped to the nuclear genome.** Left, $2 \times 151$ bp in ROSMAP and right, $2 \times 149$ bp in the lifespan WGS. The data underlying this figure can be found in S1 Data.
(PDF)

**S2 Fig. Boxplots of Numt count in matched cerebellum and DLPFC samples from $n = 9$ individuals.** $P$-value = 0.033, Student's $T$ test, paired, two-sided. Samples are shown in jittered points. The data underlying this figure can be found in S1 Data.
(PDF)

**S3 Fig. MtDNA copy number association with Numt count and age at death by tissue and cognitive impairment in ROSMAP cohort.** Correlation between mtDNA copy number and Numt count in all ROSMAP samples (A), DLPFC (B), cerebellum (C), PCC (D), whole blood (E), and PBMC (F), respectively. (G) Correlation between mtDNA copy number and age at death in all ROSMAP samples, cerebellum, PCC, whole blood, and PBMC, respectively. (H) Correlation between mtDNA copy number and age at death in DLPFC and 3 cognitive groups

in DLPFC, respectively. $r^2$ and $p$-values are calculated using standard least-squares regression models. The data underlying this figure can be found in S1 Data.
(PDF)

**S4 Fig. Cerebellum-, PCC-, and whole-blood-specific Numts are not associated with the age of death or cognitive status.** (A) Cerebellum samples correlated with age at death, stratified by cognitive diagnosis status. (B) PCC samples correlated with age at death, stratified by cognitive diagnosis status. (C) Whole-blood samples correlated with age at death, stratified by cognitive diagnosis status. Data points are colored by arbitrary age groups (see Methods) in light yellow, orange, and brown, respectively. $r^2$ and $p$-values are calculated using standard least-squares regression models.
(PDF)

**S5 Fig. Common fibroblast Numts are not abundant across the lifespan and are not associated with age.** (A) Numts shared between cell lines (donors) are not significantly correlated with aging. (B) Slopes from cell line-specific Numts and shared Numts in the lifespan model. The data underlying this figure can be found in S1 Data.
(PDF)

**S6 Fig. Background somatic SVs and MEIs during aging in primary human fibroblasts.** (A) Heatmap of slopes based on the linear regression between days cultured and the cell line-specific Numts (from Dinumt). (B) Heatmap of slopes based on the linear regression between days cultured and the cell line-specific MEIs (from MELT). (C) Heatmap of slopes based on the linear regression between days cultured and the cell line-specific SVs (from DELLY).
(PDF)

**S7 Fig. MtDNA copy number (mtDNAcn) association with Numt count and days cultured in lifespan model.** (A) Correlation between mtDNA copy number and Numt count in 3 treatment groups and *SURF1* defect group, respectively. (B) Correlation between mtDNA copy number and the number of days cells were cultured (Days cultured) in 3 treatment groups and *SURF1* defect group, respectively. $r^2$ and $p$-values are calculated using standard least-squares regression models. The data underlying this figure can be found in S1 Data.
(PDF)

**S8 Fig. Variant allele frequency of non-reference Numts in ROSMAP and lifespan study.** The Numts were categorized into germline ones (red), which overlapped with 1KG and Wei and colleagues callset and shared between tissues (ROSMAP, left) or cell lines (lifespan study, right), and potential somatic ones (green), which are tissue-specific (ROSMAP, left) or cell line-specific (lifespan study, right). The data underlying this figure can be found in S1 Data.
(PDF)

**S9 Fig. Effect of using the mean, median, or mode of the coverage on mtDNAcn estimates.** Each dot represents one of the 455 ROSMAP samples from the dorsolateral prefrontal cortex (DLPFC). The black line indicates the diagonal and the blue line represents a linear regression line. The Pearson correlation is displayed in the top left corner. Scatterplots (A–C) depict (A) autosomal coverage, (B) MT coverage, and (C) mtDNAcn using the median (x-axis) versus the mean (y-axis). Scatterplots (D–F) depict (D) autosomal coverage, (E) MT coverage, and (F) mtDNAcn using the median (x-axis) versus the mode (y-axis). The data underlying this figure can be found in S1 Data.
(PDF)

**S10 Fig. mtDNA copy number measures by qPCR and WGS are comparable in control and stressed fibroblasts during lifespan.** Three donors in each group are merged for analysis. R-squared values and *p*-values are calculated using standard linear regression models. The data underlying this figure can be found in S1 Data.
(PDF)

**S1 Data. Data underlying Figs 1B–1F, 2A, 2B, 2G, 3B, 3D and 4A–4D, S1, S2, S3, S4A–S4C, S5, S7, S8, S9A–S9F and S10A–S10D.**
(XLSX)

## Author Contributions

**Conceptualization:** Weichen Zhou, Kalpita R. Karan, Martin Picard, Ryan E. Mills.

**Data curation:** Weichen Zhou, Kalpita R. Karan, Wenjin Gu, Hans-Ulrich Klein, Gabriel Sturm, Ryan E. Mills.

**Formal analysis:** Weichen Zhou, Kalpita R. Karan, Wenjin Gu, Hans-Ulrich Klein, Gabriel Sturm.

**Funding acquisition:** Weichen Zhou, Philip L. De Jager, David A. Bennett, Michio Hirano, Martin Picard, Ryan E. Mills.

**Investigation:** Weichen Zhou, Martin Picard, Ryan E. Mills.

**Methodology:** Weichen Zhou, Kalpita R. Karan, Wenjin Gu, Hans-Ulrich Klein, Gabriel Sturm, Ryan E. Mills.

**Project administration:** Weichen Zhou, Martin Picard, Ryan E. Mills.

**Resources:** Weichen Zhou, Kalpita R. Karan, Hans-Ulrich Klein, Gabriel Sturm, Philip L. De Jager, David A. Bennett, Michio Hirano, Martin Picard, Ryan E. Mills.

**Software:** Weichen Zhou, Kalpita R. Karan, Ryan E. Mills.

**Supervision:** Martin Picard, Ryan E. Mills.

**Validation:** Weichen Zhou, Kalpita R. Karan, Hans-Ulrich Klein, Gabriel Sturm, Martin Picard.

**Visualization:** Weichen Zhou, Kalpita R. Karan, Wenjin Gu.

**Writing – original draft:** Weichen Zhou, Kalpita R. Karan, Ryan E. Mills.

**Writing – review & editing:** Weichen Zhou, Kalpita R. Karan, Wenjin Gu, Hans-Ulrich Klein, Gabriel Sturm, Philip L. De Jager, David A. Bennett, Michio Hirano, Martin Picard, Ryan E. Mills.

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
