## [Editor Report · Decision Letter 0]

13 Mar 2024

Dear Dr Mills, 

Thank you for submitting your manuscript entitled "Somatic nuclear mitochondrial DNA insertions are prevalent in the human brain and accumulate over time in fibroblasts" for consideration as a Research Article by PLOS Biology, and please accept my apologies for the delay in sending you an initial decision. I have been in and out of the office over the past two weeks with various daycare colds, and with a brief vacation - and I had wished to discuss your study with an Academic Editor with relevant expertise - which added a bit of time to the assessment process. 

Your manuscript has now been evaluated by the PLOS Biology editorial staff and by an academic editor and I am writing to let you know that we would like to send your submission out for external peer review.

Once your full submission is complete, your paper will undergo a series of checks in preparation for peer review. After your manuscript has passed the checks it will be sent out for review. To provide the metadata for your submission, please Login to Editorial Manager (https://www.editorialmanager.com/pbiology) within two working days, i.e. by Mar 15 2024 11:59PM.

Kind regards,

Luke

Lucas Smith, Ph.D.

Senior Editor

PLOS Biology

lsmith@plos.org

---

## [Decision Letter · Decision Letter 1]

12 Apr 2024

Dear Dr Mills,

Thank you for your patience while your manuscript "Somatic nuclear mitochondrial DNA insertions are prevalent in the human brain and accumulate over time in fibroblasts" was peer-reviewed at PLOS Biology. It has now been evaluated by the PLOS Biology editors, an Academic Editor with relevant expertise, and by several independent reviewers. 

In light of the reviews, which you will find at the end of this email, we would like to invite you to revise the work to thoroughly address the reviewers' reports.

As you will see below, the reviewers agree that the study is interesting. However they have provided a number of important comments aimed at improving the presentation and clarity of the findings and methods - and we think these should be carefully attended to as they are essential for readers to interpret the results and their broader implications. The reviewers also highlight that additional work will be needed to bolster and refine the analyses presented here. 

Given the extent of revision needed, we cannot make a decision about publication until we have seen the revised manuscript and your response to the reviewers' comments. Your revised manuscript is likely to be sent for further evaluation by all or a subset of the reviewers.

**IMPORTANT - SUBMITTING YOUR REVISION**

*Re-submission Checklist*

*Published Peer Review*

*PLOS Data Policy*

*Blot and Gel Data Policy*

Sincerely,

Luke

Lucas Smith, Ph.D.

Senior Editor

PLOS Biology

lsmith@plos.org

REVIEWS:

Reviewer #1, Martin Breuss (note, reviewer 1 has signed this review): In this manuscript by Zhou, Karan, and colleagues, the authors describe their analysis of somatic nuclear mitochondrial DNA insertions (Numts) in existing data from human tissues and cell lines. In line with current thinking from many laboratories interested in somatic mosaicism, the authors take a described mutational phenomenon in cancer mosaicism and interrogate its 'normal' occurrence independent of tumorigenesis. Therefore, I believe that the work is timely and interesting, and its description has value for the specific field of somatic mosaicism as well as a broader audience.

However, there are a few aspects that were unclear to me that might necessitate more explanation, different visualization of the data, or changes in their analyses.

Major concerns:

- I would like to see a bit more information on their 'somatic' definition. While I agree with their overall logic, I was missing a more detailed description of their somatic variants. Based on my understanding of the data (and their schematic in Figure 1A) it appears to me that any Numt has to be evidenced by multiple reads to obtain the length of those that go beyond one read. If that is correct, these Numts are likely clonal mosaic events evidenced in multiple cells and originating from development. Could the authors clarify this point?

o I was missing a description of the observed variant allelic fractions (VAFs) in the text or their VCFs based on how many reads evidenced an event. The only mention of VAFs I could find was in Supp. Figure 8.

o Especially a comparison to known germline events might allow the authors to define mosaicism not only by tissue-to-tissue comparison but potentially by their abundance in the data.

- As this is used quite a bit in Figure 2, I was surprised that the authors did not include a categorical comparison of the disease and age groups for Numt counts. I appreciate that some of this can be deduced from the other figures, but it is not intuitive for a reader.

o Related to age groups: The authors state that they were roughly defined by picking equally sized groups (i.e., doing a separation by thirds). However, the numbers are 295, 319, and 545. This does not appear to be defined by group size, leaving me wondering how the age groups were actually defined by the authors. 

- I am unconvinced that using PBMCs with their relatively low sample size is a good/useful baseline for comparison (e.g., Figure 1C). Similarly, it might be better to remove samples with very low sample sizes from the main analyses (as is done for 1D, for instance).

- The authors use Student's t-test for the data presented in Figures 1C and D. In my mind, this should be an ANOVA or comparable test, as they compare all groups to each other without a prior hypothesis. This is further complicated by the vast differences in sample sizes which might complicate some of the standard assumptions for these tests.

- For Figure 1E, as the calling of Numts relies on the mappability of the companion read, could the differences be explained by a systematic bias in identifying Numts in intergenic regions? I saw in the methods for lifespan some mention of GC content and repeat sequence, but I am unsure whether this was taken into account for the analysis in 1E.

- Related to my first comment and relevant for the analyses presented in Figure 2: the negative correlation for DLPFC (especially in NCI) is intriguing. However, the implications may be different depending on whether these events likely arise in development and are found in neurons/neuroglia or are unique private somatic events. Based on the blood data, I would assume that they are likely not derived from infiltrating immune cells but could still be found in microglia. Do the authors have any thoughts on this? I was surprised that the findings of Figure 2 were barely mentioned in the Discussion section.

- It is unclear to me from reading the manuscript and methods what the definition of 'cell-line(donor)-specific' vs 'whole-genome overall' Numts is. The difference appears to be small for Numts but large for the deletions. This should be better described, as I am currently not sure what to do with this information.

Minor concerns:

- I am confused by their sentence on lines 145-147: The study cited in reference 35 has a very specific study design to enrich for neocortical variants. Thus, if the authors wanted to highlight the increase in Numts in neocortex vs. the cerebellum, this study is likely not a very good benchmark.

- The variability in axes scaling (e.g., Figure 2, Figure 3 for 'lefts' and 'rights') makes it harder for the reader to compare different data.

- I am unsure why Figure 4D appears to be the only one that does not show individual data points or a box and whisker plot unlike most other figures.

- I disagree with the use of the word 'colonization' on line 293 in this context. It implies an active or selected process that benefits the colonizing entity, which I believe is not adequate here.

Reviewer #2: This is an excellent paper on a generally underappreciated phenomenon. One wonders if the fact that NUMTs are seen a "contaminants" in mitochondrial sequencing works has lead to a negative view of them and an under-appreciation that these serve as mutational fragments that can litter our nuclear genomes. In fact, NUMT insertions have been found to be the cause of (non-mitochondrial related) genetic disease (and I would suggest the authors highlight this a bit more in the discussion - see below). 

From a technical perspective, I have no real concerns. The analysis using the dinumt pathway has been robustly tested elsewhere. Their tissue analysis was quite straightforward. With the sampling limitations better matching was not possible, as clearly stated in the text (which is very appreciated). Limitations to some conclusions due to sample size are clearly stated for the readers, and this conservative, careful approach is appreciated in our current era of "over hyping" of research results. The clear limitations section in the discussions is well done, and as one who also struggles for get "the right" tissue samples for analysis, I can understand the sampling limitations the authors faced. 

My only negative comments are editorial in origin, and can be addressed without any further experimentation or analysis.

Major comments;

This paper is going to be important for non-neurologists. As such, I would suggest the brain region and other acronyms be kept to a minimum (excluding figures, where the acronym can be spelled out in the legend). Even though the acronyms are spelled out at the start of the results, it is very difficult for a non-specialist (like myself) to follow these unfamiliar short forms (DLPFC, PCC, AC, PBMC, etc). 

The only part of the manuscript I have trouble with is the discussion on page 15, where the authors discuss the fact that numts may have functional meaning (lines 329 to 348). Contained within Reference 8 of their manuscript, citations 32-37 list diseases where the genetic lesion or mutation contains a NUMT. I strongly suggest that this be entered in to this argument, as these are clear examples of a functional problem due to a numt insertion. 

Line 33 - I believe here they are talking about the population distribution of numts? If so, the evolutionary constraint infers functional consequence, and this functional consequence point should be more explicitly. Even tied into the disease cases that I mention above. 

Line 334 - I am not sure whey they are trying to link Numts moving from the mitochondria and inserting into the nucleus to mitochondrial disease. NUMTs will lead to insertions, which are potentially damaging to the site they insert into. So selection against the MELAS 3243 allele is not really relevant to nuclear DNA insertion mutations. If there are parallels to CNV variation, or other rearrangement mutations in the blood it would be relevant, but selection on heteroplasmic variants has little correlation to nuc-DNA insertions. 

Lines 344-345 - An insert of any size is likely to disrupt a protein sequence, or could disrupt a promoter, etc. So I am not clear on the logic that the size of the insert will lead to "more likely deleterious effects". Can this be spelled out more clearly or omitted?

Minor points. 

Line 33 - NUMT would be singular and NUMTs plural? So should it be Numts are…. on this line?

In the discussion of NUMTs across species, the authors appeared to mis a few well documented extreme examples, such as the extreme arrays of recent numt inserts in the domestic cat (PMID: 37820546, PMID: 7932781, PMID: 17660503) and the very high percent of the genome made up on numt sequences in the honeybee (PMID: 17383971) and the possum (https://www.frontiersin.org/articles/10.3389/fevo.2022.844443/full). 

In your analysis of new NUMTS - did there happen to be haplogoup informative sequences on the mtDNA sequence that may match the mtDNA of the person and the origions of the NUMTS in any of the reported novel spontaneous insertions?

Reviewer #3: In this paper, Zho et al. study age-related patterns in NUMT insertions in the human genome. They show provide evidence of ongoing numtogenesis (insertion of NUMTs in the nuclear genome) in somatic tissues with time. I think they present important empirical evidence supporting this that has been missing from the literature and I am overall convinced of this result if I don't think too hard about what they are doing. Therein lies my issue: The methods are sometimes unclear and the use of statistical terminology is very loose in certain places, which sometimes makes it difficult to follow exactly what they are doing. I will highlight some of these problems below:

 L147: Could mtDNA copy number estimation be affected by NUMT count? Could that lead to the association observed?

I don't understand what the percentages in L163 mean. For example, is 45%, the percentage of NUMTs that were discovered within introns? Then, how is 34.92% defined? Also, I don't understand the phrase "compared to its overall composition rate of the genome". Please be clear about the metric being analyzed.

L165-166: "significant differences in genic distribution were further observed across the various tissues". Here "genic distribution" is a nebulous term and Fig. 1E or its caption are not clear about this either. For example, the y-axis in Fig. 1E is not defined and the caption defines this as "genic distribution". I have no idea, mathematically speaking, what this is.

L166: "moderate gene tolerance and potential negative selection of somatic Numt integration in the human brain and blood cells". These two things seem contradictory and I'm not sure how this conclusion follows from the observation that introns are enriched and intergenic regions are depleted of NUMTs compared to the genomic background.

L171: "permuting the positions of our observed NUMTs 50,000 times. Permuted with respect to what? The methods weren't clear about this either.

L174: "we observed no significant deviations from random for any of the tested tissues". Significant deviations in what metric? They refer to Fig. 1F, which shows a z-score on the y-axis and the caption does not describe what this z-score represents.

L179: "paucity of Numts overall may be underpowered" - power is defined for a statistical test. I think they mean to say that the paucity of NUMTS may lead to their permutation test being underpowered?

L187: "each tissue in aggregate" - presumably they mean analyzing all NUMTs in aggregate across tissues.

L201-202: "pathogenicity of AD is likely uncoupled from age-dependent Numt integration". I think this conclusion is coming from the observation that the correlation between NUMT count and age was not statistically significant in AD patients but was significant in the other cohorts. But I don't think they can make this claim without formally testing the slopes across cohorts. The absence of significance in AD patients could be because of lack of power. It does not suggest that there is no age-dependent NUMT integration in AD patients.

L245: "somatic Numts could be driven by the simple clonal expansion of few Numts-containing cells". This confused me because throughout the paper I've been thinking of the response variable as being the number of NUMTs observed, not their frequency (depth of NUMT containing reads/total depth at insertion site). I can see how clonal expansion might increase the frequency of existing NUMTs but not how it would lead to an increase in the number of NUMTs.

L519: "Normalized the NUMT count". It sounds like this is done for each individual. What was the normalization?

L520: "Values of Numt Numbers were normalized by the median of Numt counts in each category." This further confused me. Is this referring to the individual-specific normalization discussed in L519 or an additional step?

L532: Why was 10Mb chosen to bin the genome? 

L548-L550: "test the significant difference between Numt genic distribution and reference genic distribution" - What is the exact metric being compared? What is "reference genic distribution"? Does "reference" imply genome background/non-genic regions?

Other questions: 

What commands were used to generate autosomal and mtDNA coverage? Samtools? Was any filtering involved?

In conclusion, the paper adds value to the scientific literature, but the authors need to be clearer about their methods.

---

## [Decision Letter · Decision Letter 2]

12 Jun 2024

Dear Dr Mills,

Thank you for your patience while we considered your revised manuscript "Somatic nuclear mitochondrial DNA insertions are prevalent in the human brain and accumulate over time in fibroblasts" as a Research Article at PLOS Biology. Your revised study has now been evaluated by the PLOS Biology editors, the Academic Editor and the original reviewers.

As you will see below, the reviewers are in agreement that the revision has addressed most of the points from the last round of review, and both reviewers 1 and 2 are fully satisfied. However, reviewer 3 has a few last suggestions to strengthen the study further and we think these should be addressed (largely with textual changes). In light of these reviews, we are pleased to offer you the opportunity to address the remaining points from reviewer 3 in a revision that we anticipate should not take you very long. We will then assess your revised manuscript and your response to the reviewers' comments with our Academic Editor aiming to avoid further rounds of peer-review, although might need to consult with the reviewers, depending on the nature of the revisions.

**IMPORTANT: In addition to addressing the reviewer requests, below, we also ask that you attend to the following editorial requests, detailed here: 

1) TITLE: After some discussion within the editorial team, we are wondering if the title could be strengthened a bit, as this might be more enticing to our broad readership. 

If you agree, and if supported, we would suggest you change the title to something like "Nuclear mitochondrial DNA insertions occur spontaneously in somatic tissues and accumulate in the human brain throughout life". 

We are happy for you to refine this further. 

2) ETHICS STATEMENT: Thank you for providing an ethics statement in the methods section of your manuscript. Can you please update this to include an approval number for the protocol approved by the Rush University IRB? And please specify which specific IRB provided approval for protocol #AAAB0483. As a last request, please indicate whether the human studies were conducted according to the principles expressed in the Declaration of Helsinki. 

3) DATA: Thank you for providing a data availability statement in our online system, detailing where the various datasets were obtained from. 

>>For data from third parties (ex the data from NIAGADS and the 1000 genomes project), please add contact information for these organizations, where readers could apply to gain access to the data. For more details on third party data sharing requirements, please see our data availability policy: https://journals.plos.org/plosbiology/s/data-availability#loc-data-management-plans

>> For the fibroblast genome sequencing data from (Sturm et al 2022), please point readers to a publicly available repository containing that data. 

>> For the callsets provided on github, please generate a DOI for this dataset to make a permanent record. See the process for doing this here: https://docs.github.com/en/repositories/archiving-a-github-repository/referencing-and-citing-content

4) DATA: In addition to the datasets referenced in your data availability statement, we also ask that you provide the individual quantitative observations that underlie the data summarized in the figures as a supplementary excel file. Please ensure that all data files are uploaded as 'Supporting Information' and are invariably referred to (in the manuscript, figure legends, and the Description field when uploading your files) using the following format verbatim: S1 Data, S2 Data, etc. Multiple panels of a single or even several figures can be included as multiple sheets in one excel file that is saved using exactly the following convention: S1_Data.xlsx (using an underscore).

>>Regardless of the method selected, please ensure that you provide the individual numerical values that underlie the summary data displayed in the following figure panels as they are essential for readers to assess your analysis and to reproduce it:

Fig 1B-G; Fig 2A-B; Fig 3 B-D; Fig 4A-D;

Fig S1; Fig S2; Fig S3A-H; Fig S4A-C; FigS5A-B; Fig S7A-B; Fig S8; Fig S9A-F; Fig S10A-D;

>>Please also ensure that figure legends in your manuscript include information on where the underlying data can be found, and ensure your supplemental data file/s has a legend.

>>Please ensure that your Data Statement in the submission system accurately describes where your data can be found.

5) CODE: Per journal policy, if you have generated any custom code during the course of this investigation, please make it available without restrictions upon publication. Please ensure that the code is sufficiently well documented and reusable, and that your Data Statement in the Editorial Manager submission system accurately describes where your code can be found.

**IMPORTANT - SUBMITTING YOUR REVISION**

*Resubmission Checklist*

*Published Peer Review*

*Blot and Gel Data Policy*

Sincerely,

Luke

Lucas Smith, Ph.D.

Senior Editor

PLOS Biology

lsmith@plos.org

REVIEWS:

Reviewer #1, Martin Breuss (note, reviewer 1 has signed this review): The authors have addressed all my concerns and no additional major concerns arose from their response to other reivewers.

Reviewer #2: All of the comments I made are addressed to my satisfaction. They reply to point 8, while not explicitly publishable, is very encouraging that the observed treands in the NUMTs are true novel insertions. I appeciate that the authors did this, and replied so candiidly. 

Reviewer #3: I thank the authors for approaching the comments constructively and taking the time to respond. I have some minor suggestions: 

In response to point 2 of mine, the authors have described clearly what the different percentages refer to: "the proportion of our detected NUMTs that fell within each region". This is more explicit than the "rate of NUMT insertion" as phrased in the main text and I encourage them to use the former language for clarity.

Re: point 4, the revised text is still vague and not saying much. Presumably, the evidence for negative selection in a particular tissue would be a depletion of NUMT insertion into coding or UTR regions compared to background expectation. This was not observed (Fig. 1F). Instead there were significant differences in NUMT enrichment in intronic and intergenic regions that the authors interpret to be due to mapping artifacts. Yet they continue on to say that "consistent significant differences in genic distribution were further observed across the various tissues, suggesting a potential negative selection of somatic NUMT integration in the human brain and blood cells" in Line 179 of the revised text. How are they able to draw this conclusion? Which aspect of the genic distribution are they referring to? What direction of difference is expected in the presence of negative selection? Do they see that?

Re: point 8, the revised text in the response document is correct but in Line 200 of the revised text, the authors still say "We first examined each tissue in aggregate and observed almost no correlation in NUMT abundance with the age of death". They mean to say: "We first examined NUMTs in aggregate across tissues observed no correlation in NUMT abundance with the age of death".

In Line 183 of the revised text, the authors say they hypothesized that the genomic distribution of somatic NUMT integration sires may differen between tissues or across age groups. They tested this hypothesis by permuting NUMT positions 50,000 times across the genome and assessing

---

## [Editor Report · Decision Letter 3]

26 Jun 2024

Dear Dr Mills,

Thank you for the submission of your revised Research Article "Somatic nuclear mitochondrial DNA insertions are prevalent in the human brain and accumulate over time in fibroblasts" for publication in PLOS Biology and thank you for addressing the last reviewer and editorial requests in this revision. On behalf of my colleagues and the Academic Editor, Thomas B.L. Kirkwood, I am pleased to say that we can in principle accept your manuscript for publication, provided you address any remaining formatting and reporting issues. These will be detailed in an email you should receive within 2-3 business days from our colleagues in the journal operations team; no action is required from you until then. Please note that we will not be able to formally accept your manuscript and schedule it for publication until you have completed any requested changes.

**IMPORTANT: A few last editorial notes and requests**

1) TITLE: Thank you for updating your title, in response to our editorial request. Given your feedback on our title suggestion, and after discussing this further with the Academic Editor, we actually end up preferring your original title (the Academic Editor agreed with you that we overstepped a bit with our suggestion). I have therefore gone ahead and reverted the title to your original ("Somatic nuclear mitochondrial DNA insertions are prevalent in the human brain and accumulate over time in fibroblasts"), in our editorial manager system - and if you agree with that change, please update your manuscript file accordingly. 

Sorry for flip-flopping on this point and happy to discuss this further if helpful. 

2) DATA AVAILABILITY STATEMENT: Thank you for updating your data availability and code availability statements in your response to editorial requests. As an FYI - I have copied and pasted this into the relevant section of our online system. Please do take a quick look at the 'data availability statement' section in our editorial manager system to make sure everything looks OK with this change. 

3) DATA: Thanks also for providing an S1_data file with the numerical data underlying your figures. Please add a sentence to each figure legend directing readers to this file. For example, you can add the sentence "the data underlying this figure can be found in S1_data". 

PRESS

Sincerely, 

Luke

Lucas Smith, Ph.D.

Senior Editor

PLOS Biology

lsmith@plos.org